# The Highly Leukotoxic JP2 Genotype of *Aggregatibacter actinomycetemcomitans* Is Present in the Population of the West African Island, Sal in Cape Verde: A Pilot Study

**DOI:** 10.3390/pathogens11050577

**Published:** 2022-05-13

**Authors:** Johannes J. De Soet, Rolf Claesson, Dorte Haubek, Anders Johansson, Mark J. Buijs, Catherine M. C. Volgenant

**Affiliations:** 1Academic Centre for Dentistry Amsterdam (ACTA), Department of Preventive Dentistry, Vrije Universiteit Amsterdam and Universiteit van Amsterdam, 1081 LA Amsterdam, The Netherlands; jjdesoet@gmail.com (J.J.D.S.); m.buijs@acta.nl (M.J.B.); 2Department of Odontology, Umeå University, 901 87 Umeå, Sweden; rolf.claesson@umu.se (R.C.); anders.p.johansson@umu.se (A.J.); 3Section for Paediatric Dentistry, Department of Dentistry and Oral Health, Aarhus University, 8000 Aarhus, Denmark; dorte.haubek@outlook.dk

**Keywords:** JP2 genotype, leukotoxin, *Aggregatibacter actinomycetemcomitans*, Cape Verde, geographic dissemination

## Abstract

*Aggregatibacter actinomycetemcomitans* is strongly associated with severe periodontitis, possibly due to its production of a potent leukotoxin. A genetic variant, the JP2 genotype, was found to produce more leukotoxin than the wild type because of a mutation in the leukotoxin gene, and this genotype is frequently found in African populations. The aim of this study was to investigate whether this JP2 genotype can be found in a randomly selected group of inhabitants of Sal, Cape Verde. Twenty-nine adults between 20 and 59 years of age (58.6% female) participated, and information on their oral health and living standards was collected. An oral examination was performed for each participant, including DMF-T and CPI scores. Plaque and saliva samples were collected and transported to Europe, where DNA was isolated, and the concentration of *A. actinomycetemcomitans* and its JP2 genotype was determined using dedicated PCR analyses. All 29 plaque and 31% of the saliva samples harboured *A. actinomycetemcomitans,* and two participants were positive for the JP2 genotype. The presence of this JP2 genotype was not associated with either CPI or DMF-T. This pilot study is the first to describe the presence of the *A. actinomycetemcomitans* JP2 genotype in a Cape Verdean population living in the Cape Verde Islands, and the findings warrant further research.

## 1. Introduction

Oral biofilm in humans is the prime initiator of inflammatory reactions in the oral cavity [1]. Microorganisms in these biofilms can produce toxic compounds that negatively affect the dental tissues, and inflammation can be initiated by activating the immune response of the host, which altogether may result in periodontitis [2]. Several so-called keystone pathogens are associated with periodontitis, among which *Aggregatibacter actinomycetemcomitans* is the most prominent species related to localized aggressive periodontitis [3,4]. This is considered to be because it possesses specific virulence factors involved in the induction of pro-inflammatory cytokines. The most prominent *A. actinomycetemcomitans* virulence factor is leukotoxin, which is cytotoxic to immune cells and induces the release of interleukin 1β [5,6].

The gene responsible for leukotoxin production has been studied intensely, and it was found that a certain genotype of *A. actinomycetemcomitans* contains a variant with a deletion in the promoter region of this gene. This results in an upregulation of the gene and a 10- to 20-fold higher production of leukotoxin compared to other strains of *A. actinomycetemcomitans* [7,8,9]. This genotype, JP2, is named after a strain that was isolated from an 8-year-old boy with deep periodontal pockets [7]. The JP2 genotype has an estimated origin in the Mediterranean region of Africa and has spread to other parts of the world following human migration routes [10,11]. Furthermore, a microevolutionary study carried out on a collection of *A. actinomycetemcomitans* strains, including from members of families originating from Cape Verde but living in Sweden, revealed colonization by the JP2 genotype [10]. It is therefore hypothesized that the JP2 genotype of *A. actinomycetemcomitans* may be present, and even rather prevalent, in the population living on the West African island of Cape Verde.

The clinical signs of periodontitis, whether or not they are carriers of *A. actinomycetemcomitans*, are bleeding on probing and periodontal attachment loss, and the presence of the JP2 genotype and the clinical signs were previously reported in adolescents in Africa, especially in the north and west [10,12,13,14,15]. Immigrants from Cape Verde to Northern European countries were also found to be carriers of the JP2 genotype, and it could therefore be expected to be found in Cape Verde [10,16].

Cape Verde consists of 11 islands (Appendix A) and is located approximately 600 km northwest of the African coast (Appendix A). Its inhabitants have a strong genetic relationship with both Central and West Africa and with Portugal, and so the prevalence of the *A. actinomycetemcomitans* JP2 genotype in Cape Verde might be expected to be higher compared to the prevalence in European and American populations (1.2% in Sweden) [17]. The aim of this study was to investigate whether the JP2 genotype can be found in a randomly selected population from the island of Sal in Cape Verde.

## 2. Results

### 2.1. General Results

A summary of the clinical and microbiological data is shown in Table 1. In total, 29 Sal inhabitants participated in the present study. The average age was 39.2 years. Of the 29 participants, 24% had completed secondary school, and 48% reported that they could read and write. In addition, 26% smoked (any type), 94% performed daily toothbrushing, and 48% reported frequent oral pain. Of the 29 participants, 17 had at least one periodontal pocket at least 6 mm deep, 9 had at least one of 4–5 mm, and 3 had none. Only two participants (7%) had no gingival bleeding after probing the index teeth.

*A. actinomycetemcomitans* was found in all 29 plaque samples, and the concentration had a mean log CFU/sample of 4.4 ± 0.67. *A. actinomycetemcomitans* was found in nine (31%) saliva samples with a mean log CFU/mL 3.0 ± 0.50. From these individuals, two periodontal samples were positive for the JP2 genotype, one in the plaque and one in the saliva of different people. This indicates a carriage rate of 6.9%. The presence of the JP2 genotype was not associated with any of the measured clinical data (gingival bleeding, CPI-index, and DMF-T; *p* > 0.05, Chi-Square test). The number of positive JP2 cases was low.

### 2.2. Carriers of the JP2 Genotype of A. actinomycetemcomitans

The first JP2-positive patient (Patient 1) was a 20-year-old female who was unrelated to any of the other participants in the study. She reported being a non-smoker and having pain in her mouth (DMF-T = 7; bleeding from the periodontal pockets around all the index teeth; periodontal pocket depth not higher than 4–5 mm). The JP2 genotype of *A. actinomycetemcomitans* was found in the saliva sample of Patient 1 only. This finding was confirmed through separation in an agarose gel (Figure 1). The total amount of *A. actinomycetemcomitans* in the patient’s saliva and plaque samples was 3.0 log CFU/mL and 4.6 log CFU/sample, respectively.

The second JP2 genotype-positive patient, Patient 2, was a 39-year-old female, also unrelated to any other participant. It is not known if she was a smoker or had oral pain. DMF-T was 3, although this could have been higher since only 14 teeth were present. The exact reason for the absence of several teeth was not documented but was most likely because of periodontal tooth loss, hence the low DMF-T value. The remaining teeth showed bleeding on probing, and the highest measured pocket depth was 5 mm (with recessions of the gums). The JP2 genotype of *A. actinomycetemcomitans* was only found in Patient 2′s plaque sample, collected from the periodontal pocket of tooth 25, which had the deepest periodontal pockets. The total amount of *A. actinomycetemcomitans* in the saliva and plaque samples was 3.3 log CFU/mL and 4.5 log CFU/sample, respectively.

## 3. Discussion

In this paper, we report the presence of the high leukotoxin-producing JP2 genotype of *A. actinomycetemcomitans* in adults living on Sal island in Cape Verde. Based on the finding of the JP2 in two (6.9%) unrelated individuals on the island in this small sample (n = 29), it is likely that more individuals on Sal are positive for this particular genotype. This study is clearly a pilot since the number of participants is low, but it is nevertheless the first study of the presence of the *A. actinomycetemcomitans* JP2 genotype in this part of Africa.

In previous studies, low JP2 prevalence (0.7%) was found in Kenya [19], with higher levels in other African countries, including Morocco (7.3%) and Ghana (8.8%) [12,15]. Earlier studies reported a strong association between the presence of the JP2 genotype and periodontitis [6,20], while other studies did not find such a clear association [15]. In the present study, the prevalence of the *A. actinomycetemcomitans* JP2 genotype was not associated with any other parameters, which is most probably due to the low number of participants involved. However, it should be noted that all participants carried *A. actinomycetemcomitans* in their dental plaque but not in their saliva in detectable amounts. This may be due to lower concentrations of the bacteria in the saliva samples due to dilution. Moreover, it is quite remarkable that the JP2 genotype was found in either the plaque or the saliva of the positive patients. When a saliva sample is positive, it might be expected that the plaque sample would also be positive, but this was not the case, suggesting that the JP2 genotype had its habitat in another tooth than the one that was sampled.

The result of this pilot study suggests a larger-scale examination would be of interest to understand the prevalence of the JP2 genotype in adolescents as well as the adult population across the island group (Appendix A). This would develop more comprehensive and detailed information about *A. actinomycetemcomitans* JP2 genotype carriers and periodontitis in the Cape Verdean population and also provide insight into the clonal diversity of the JP2 genotype found in places as remote and isolated as Cape Verde. Such a study would need careful preparation since resources on Cape Verde are scarce, and all materials would have to be imported. There are virtually no facilities that could be classified as microbiological laboratories or dental clinics. In terms of samples, our results suggest that several teeth should be examined individually and more oral sites used per participant to ensure a good representation of the oral flora. Possibly, 16S sequencing could be considered to gain insight into the ecosystems in which the JP2 genotype resides. We cannot say anything about the associations between oral health, toothbrushing or educational level for the participants in this study, but it was observed that the individuals carrying the JP2 genotype did have tooth decay and extensive gingival bleeding.

Based on the demographic, local, logistical, clinical, and microbiological knowledge obtained in this pilot study, a more comprehensive and standardized recruitment design and set-up are required. Moreover, participants from Cape Verdean islands other than Sal should be included, as the local environment may influence findings in different populations. Furthermore, clinical parameters must be chosen that clearly distinguish between gingivitis and periodontitis but are also ethical to measure in a population with little dental care. However, the main conclusions of the present pilot study are:The prevalence of the JP2 genotype of *A. actinomycetemcomitans* is higher on the Cape Verdean island of Sal than in Europe;The prevalence of this JP2 genotype could not be associated with the presence of periodontitis or gingivitis, as measured by CPI, because of the relatively low sample size;These results warrant the initiation of a larger study on the prevalence of this potentially more harmful *A. actinomycetemcomitans* genotype.

## 4. Materials and Methods

Inhabitants of Sal were invited to present at a simple dental clinic and to bring relatives, if possible. After reading an information letter and signing informed consent, individuals aged 18 years or older were included in the study. A native speaker from Sal read the information letter to people who were illiterate.

All participants were asked to provide information about their age, gender, dental pain, and tooth brushing habits, and a brief questionnaire about socio-economic status was also used. A translator, fluent in the Cape Verdean language, English, and Dutch, was present to assist participants with filling out the questionnaires since many inhabitants of Sal are illiterate. During clinical examination, the number of decayed, missed, and filled teeth (DMF-T) was assessed using the ICDAS score of three as the cut-off level for dental caries. DMF-T was only recorded for teeth with clear evidence of caries, and missing teeth without a known history of caries, trauma, or periodontitis were not recorded as missing in DMF-T. Periodontal health was assessed using the Community Periodontal Index (CPI), focusing on gingival bleeding and the periodontal pockets of 10 index teeth with molars paired 17/16, 11, 26/27 and 47/46, 31, 36/37 [21].

After the periodontal screening, a subgingival dental plaque sample was collected from the four mesial sites of the first permanent molars using a curette. In cases where one or more first permanent molars were absent, an adjacent molar or premolar in the same quadrant was sampled. Unstimulated saliva was sampled from under the tongue (left and right side) using a cotton swab [22]. The samples were put in a sterile tube with RNAprotect^®^ (Qiagen, Venlo, The Netherlands) for fixation [23]. The samples were immediately stored at 4 °C and, within four days of sampling, transported to The Netherlands on ice. After arrival at ACTA, the samples were centrifuged, and the RNAprotect^®^ was removed. The pellets were resuspended in 100 µL sterile Tris-EDTA buffer and added to wells of a 96-deep-well plate containing Tris-saturated phenol, 0.1 mm zirconium beads, and lysis buffer. Samples were mechanically lysed by bead-beating at 1200 rpm for 2 min, and DNA was purified with the Mag MiniKit (LGC Genomics, Berlin, Germany) [24].

The DNA samples were then sent to the Department of Odontology, Umeå, Sweden, for qPCR-based quantification of *A. actinomycetemcomitans* as described earlier by Ennibi [25]. Briefly, the total concentration of *A. actinomycetemcomitans* in the isolated DNA samples was determined using primers specific for the *LtxA* gene (forward: CTAGGTATTGCGAAACAATTTG; reversed: CCTGAAATTAAGCTGGTAATC). Cycle settings for the *LtxA* primers were held at 95° for 10 min, 45 cycles of 95° for 10 s, and 55° for 5 s. The quantification of the JP2 genotype was based on primers and probes specific for JP2 genotype-associated DNA sequences within the leukotoxin promoter region (forward: TCTATGAATACTGGAAACTTGTTCAGAAT, reversed: GAATAAGATAACCAAACCACAATATCC, probe: FAM-ACAAATCGTTGGCATTCTCGGCGAA-TAMRA). Cycle settings for the JP2 primers were held at 95° for 10 min, 45 cycles of 95° for 10 s, and 58° for 5 s. To quantify both the total *A. actinomycetemcomitans* and the JP2 genotype concentration, standard curves with 10^1^–10^8^ cells/mL were prepared for strain HK1651. In addition, selected samples from the qPCR- quantification of the JP2-specific gene were separated in an agarose gel (Figure 1).

Statistical analysis of the associations between the presence of the target bacteria and clinical parameters was performed using a chi-square test

The study protocol was evaluated and approved by the Institutional Review Board of the Vrije Universiteit Medical Center Amsterdam (reference number 2016449) and was subsequently approved by the Ministry of Health of Cape Verde.

## Figures and Tables

**Figure 1 pathogens-11-00577-f001:**
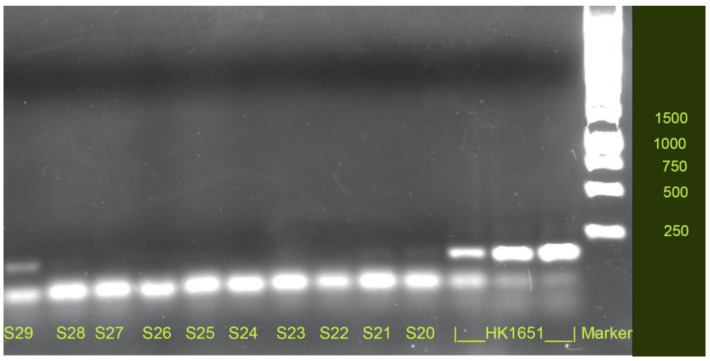
Confirmation of the JP2 genotype in saliva. Amplified DNA from qPCR-based quantification of the JP2 genotype was separated on an agarose gel using saliva from the JP2-genotype positive patient (S29) and from 10 negative samples. DNA from HK 1651 [18] was used as a positive control. The JP2-genotype is clearly visible on the basis of one extra band, around 200 base pairs.

**Table 1 pathogens-11-00577-t001:** Demographical, clinical, and microbiological data of the study population.

		SD	Range	N
Gender (female)	58.6%			29
Age (years, mean)	39.2	12.6	20–59	27
Smoking habits (% positive)	26.1%			24
Oral hygiene (% applied)	58.6%			17
Teeth present (mean)	23.1	6.7	8–31	29
Teeth with caries (mean)	3.7	3.2	0–10	29
CPI (median)	2.5		1–4	29
Bleeding on Probing	92.9%			29
Aa saliva (median Log CFU/mL)	4.7		2.7–3.8	29
Aa plaque (median Log CFU/sample)	3.0		3.6–6.0	29

The most relevant demographical and clinical data of the study population, given in percentages of the population, median with range or mean with standard deviation, where applicable. Teeth with caries do not include teeth that may be missing because of caries. Aa refers to the concentration of *A. actinomycetemcomitans* in the Log CFU/mL sample.

## Data Availability

The data presented in this study are openly available as supplementary material to this publication.

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
