# Peer review of "The Highly Leukotoxic JP2 Genotype of Aggregatibacter actinomycetemcomitans Is Present in the Population of the West African Island, Sal in Cape Verde: A Pilot Study"

_pathogens, 2022, doi:10.3390/pathogens11050577_

Round 1
Reviewer 1 Report
In this study the presence of Aggregatibacter actinomycetemcomitans and the JP2 genotype in subgingival plaque samples and saliva samples was examined. In this pilot study at the island of Sal, Cape Verde, 29 individuals were partially examined, and samples were harvested. All individuals examined had traceable amounts of A. actinomycetemcomitans in plaque and 31 % in saliva. Two subjects had the JP2 genotype, one detected in saliva and one in plaque sample. The authors reported no association between the presence of JP2 genotype and the clinical parameters recorded.
This is an interesting study evaluating the prevalence of the JP2 genotype at Cape Verde, especially with regard to previous findings of immigrants from Cape Verde colonized with this genotype. There are, however, limitations that need to be addressed:
INTRODUCTION:
In the introduction the authors write that “the clinical signs and symptoms of carriers of the JP2….is bleeding on probing and periodontal attachment loss” which may be interpreted as it is the JP2 genotype that solely causes the BoP and attachment loss. These clinical findings are used to diagnose patients with periodontitis irrespective of the presence or absence of A. actinomycetemcomitans.
-Why was only A. actinomycetemcomitans examined and not other “putative pathogens”?
- There is no need for Figure 1 and Figure 3
M&M
- Since this is a pilot study it is understood why there is a limitation in the number of study participants. There are, however, no evident reasons why the authors chose to use the The Community Periodontal Index (CPI) instead of a full-mouth periodontal examination of the periodontal status. Although, CPI has previously been used in many epidemiological studies and surveys to estimate the periodontal status of populations due to its simplicity and speed the index is limited in estimating the true extent and severity of disease. Since A. actinomycetemcomitans is associated with periodontitis and not caries, the recordings of attachment loss would have added more value than recordings of the presence of decayed teeth.
- How were the subgingival samples harvested, using paperpoints or some other method?
- The methods used, reference 22-25 should be described briefly in the text
- The authors should explain what statistics they used
RESULTS
- A table of the characteristics of the examined subjects is missing as is data of DMFT, CPI, gingival bleeding, etc. within the examined population. How sick was this population?
- DMFT includes decayed, filled and, missing teeth. How can patient number 2 have a DMFT of 3 but have only 14 teeth left?
- CFU/ml is reported. Were the samples cultured and colony forming units counted?
DISCUSSION
- There is a discussion missing on CPI and the lack of association between the presence of JP2 genotype and periodontal status. This was the aim of the study?
- Why is there a need of a new and larger study? What do you expect to find in the larger study? Why do you want to publish this study if you intend to perform a new but larger one?
- In one of the subjects the JP2 genotype was found in saliva but not in plaque. Is this person a carrier of this genotype or is it possible that this genotype is one of the transient flora?
Author Response
We thank the reviewer for the careful work and comments on our manuscript. The detailed reply from us is attached in the MS word document

Reviewer 2 Report
This is a well written study report on the detection of A.a. JP2 clone in a population living at the Cape Verde island.
I will have only three minor comments:
1) Plaque levels (CFU) of JP2 are given for 2 JP2 positive participants. For others Aa levels are given as log CFU. That makes the comparison difficult.
2) Please write the bacteria names in italics (see results section)
3) Paper mentions about a frequent type of pain in the mouth. Can that be described a bit more (toothache or of mucosal origin?)
Author Response

(The authors gave the same response as above.)

Reviewer 3 Report
Comments to Authors
This manuscript reported the prevalence of JP2genotypes for A. actinomycetemcomitans in the population living on the West African Island, Sal, at Cape Verde. These findings were fundamentally interesting for understanding the highly leukotoxic genotype spreading through human migration. However, this manuscript has some problems from the scientific point of view.
- In Figure 2, the authors did not describe the JP-2 positive sample (perhaps S29). They should show it by a symbol or explain it.
- In lines 113-122, the authors described another JP-2 positive sample. They should show the agarose gel electrophoresis data as shown in Figure 2.
- In this manuscript, the bacteria name A. actinomycetemcomitans is not Italic in many lines (lines 20, 89, 93, etc.) Check all the bacteria names.
Author Response

(The authors gave the same response as above.)

Round 2
Reviewer 1 Report
Additional comments
The manuscript examining the presence of A.a. and the JP2 genotype in Cape Verde has improved and below are additional comments that need to be addressed.
- The authors write that the aim of the study was to “investigate whether the JP2 genotype can be found in a randomly selected population from the island of Sal in Cape Verde”. Is this the only aim of the study or was the aim also to investigate for a possible correlation between the presence of JP2 and periodontal status. If this was not the aim, then there was no reason to examine the patients using CPI, and if this was the reason then there is a need to report you findings of the periodontal status.
- Figure 1 is still considered redundant since it is not part of any findings of the conducted experiment. Readers not knowing the location of a country can find this information elsewhere. It should be replaced with a table. A table providing information of the patient data is considered an important part in clinical studies and should be incorporated in the main text (not as a supplementary). The table should provide a summary of patient characteristics and finding, not detailed information of each participant as it is now. It is suggested that following data is presented:
- gender %,
- smoking habits %,
- Oral hygiene habits (%),
- Age -average age and age range-,
- Missing teeth or remaining teeth - average and range- (NOT DMFT since caries is not of importance in this study),
- CPI - average and range- (if you do not have more detailed information of BoP% and range, and CAL)
- At least one PPD of 6 mm - %-,
- detection of A.a (counts/ml or cells/ml, NOT CFU/ml since this is colony forming units mainly used when culturing bacteria),
- presence of JP2 genotype.
By including this data in a table, the text providing this information in results section can be shortened.
- If two subjects of 29 had the JP2 genotype then this is 6.9%, not 5.7%?
- DMFT officially not reporting all missing teeth (except third molars) in not something that is known in this part of the world and the authors should adjust their text so it may be understood internationally. It appears that these subjects have caries as well and it can not be excluded that the teeth were lost due to caries. According to the scientific literature the main cause of loss of teeth among younger individuals (< 50 years of age) in general is caries and not periodontitis.
- Regarding the readability of the study, the same goes with DPSI, you have it in your table but not in your text. I guess you are referring to CPI? CPI goes from 0-4 but you report as high recording as 5?
- This is not the first study performed in remote areas of the world lacking facilities and periodontal examinations are possible to perform in daylight using a hand mirror and periodontal probe as is the possibility to sample the oral sites, which previous studies have proven. The difference between gingivitis and periodontitis is attachment loss and that is something that is not registered using CPI, therefore you can not distinguish periodontitis patients with patients not having periodontitis. The recordings you have made are based on BoP and PPD and therefore the distinction between gingivitis and periodontitis can not be made.
- If now the authors believe that the main reason for loss of teeth in this population is indeed periodontitis, then the importance of the findings of JP2 are perhaps not that interesting and do not validate a larger study since only two subjects were positive for JP2 but 7 had less than 20 teeth and no correlation or trend to correlation was found between JP2 and oral status?
